# The Effect of Maternal Exposure to Air Pollutants and Heavy Metals during Pregnancy on the Risk of Neurological Disorders Using the National Health Insurance Claims Data of South Korea

**DOI:** 10.3390/medicina59050951

**Published:** 2023-05-15

**Authors:** Kuen Su Lee, Won Kee Min, Yoon Ji Choi, Sejong Jin, Kyu Hee Park, Suhyun Kim

**Affiliations:** 1Department of Anesthesiology and Pain Medicine, Eulji University Uijeongbu Eulji Medical Center, Eulji University School of Medicine, Uijeongbu 11759, Republic of Korea; dlrmstnx@eulji.ac.kr; 2Department of Anesthesiology and Pain Medicine, Korea University Ansan Hospital, Korea University College of Medicine, Ansan 15355, Republic of Korea; 3Department of Neuroscience, Korea University College of Medicine, Seoul 02841, Republic of Korea; holicer@korea.ac.kr; 4Department of Pediatrics, Korea University Ansan Hospital, Ansan 15355, Republic of Korea; czrabbit@korea.ac.kr; 5Department of Biomedical Sciences, College of Medicine, Korea University, Seoul 02841, Republic of Korea; dieslunae@korea.ac.kr

**Keywords:** air pollution, autism spectrum disorder, epilepsy, heavy metals

## Abstract

The objective of this study was to evaluate the effects of high levels of maternal exposure to ambient air pollution and heavy metals on risks of autism spectrum disorder (ASD) and epilepsy using the National Health Insurance claims data of South Korea. The data of mothers and their newborns from 2016 to 2018 provided by the National Health Insurance Service were used (*n* = 843,134). Data on exposure to ambient air pollutants (PM2.5, CO, SO_2_, NO_2_, and O_3_) and heavy metals (Pb, Cd, Cr, Cu, Mn, Fe, Ni, and As) during pregnancy were matched based on the mother’s National Health Insurance registration area. SO_2_ (OR: 2.723, 95% CI: 1.971–3.761) and Pb (OR: 1.063, 95% CI: 1.019–1.11) were more closely associated with the incidence of ASD when infants were exposed to them in the third trimester of pregnancy. Pb (OR: 1.109, 95% CI: 1.043–1.179) in the first trimester of pregnancy and Cd (OR: 2.193, 95% CI: 1.074–4.477) in the third trimester of pregnancy were associated with the incidence of epilepsy. Thus, exposure to SO_2_, NO_2_, and Pb during pregnancy could affect the development of a neurologic disorder based on the timing of exposure, suggesting a relationship with fetal development. However, further research is needed.

## 1. Introduction

Air pollution is a major risk factor for global health. Exposure to air pollution has been linked to increased mortality and morbidity, contributing significantly to the overall global disease burden [1]. Globally, the number of all-cause deaths from overall air pollution increased by 2.62% from 1990 to 2019 [2]. The Health Effects Institute’s State of Global Air reported that PM2.5, a type of fine-particulate air pollution, is the sixth-highest risk factor for death globally, accounting for approximately four million deaths in 2019 alone [3]. Over 90% of the world’s population live in areas where the air quality standards set by the WHO are not met. In Asian megacities, air pollution concentrations have been observed to be the highest worldwide. Recently, Korea has undergone rapid economic growth, and the air quality has worsened.

Air pollution is caused by complex components including nitrogen dioxide (NO_2_), sulfur dioxide (SO_2_), and particulate matter (PM). NO_2_, SO_2_, PM, and indirectly generated ozone (O_3_) are major air pollutants that are related to exhausts from vehicles and industrial energy consumption, which are caused by urbanization and industrialization [4,5,6,7].

Generally, heavy metals are amalgamated with PM [8] and mainly originate from diesel and gasoline exhaust fumes from local traffic and industrial areas [9,10].

Air pollution is known to affect mortality and the prognosis of cardiovascular disease, respiratory disease, and stroke [11,12,13]. Recent studies have focused on the potential effects of exposure to NO_2_, SO_2_, and PM2.5, which showed that a maternal exposure factor that can adversely affect even the prenatal period [14,15].

PM2.5, with various deleterious components, can enter the blood circulation through the lungs [16,17]. In addition, PM2.5 exposure during pregnancy can induce oxidative stress and an inflammatory response. It can affect the fetus through changes in the uterine environment and placental function [18,19]. The prenatal period is critical for brain development. It is a complicated process determined by both genetic and external factors. Deleterious factors during the prenatal period might have severe and long-term adverse effects on brain structure and function, resulting in neurodevelopmental disorders.

Some studies have shown that exposure to NO_2_, SO_2_, and PM2.5 is a risk factor for stillbirth and spontaneous abortion, supporting the notion that exposure to NO_2_, SO_2_, and PM2.5 can affect the fetus [20,21]. Regarding neurodevelopment, it is known that PM2.5 can induce oxidative stress and an inflammatory response [8] and that both oxidative stress and an inflammatory response can affect the expression of brain-derived neurotrophic factor (BDNF) and cyclic AMP response element-binding protein (CREB), which are well-known neurodevelopment factors [22,23,24]. Some studies have demonstrated that exposure to PM2.5 is related to changes in the expression of BDNF and CREB [25,26,27].

The etiology of autism spectrum disorder (ASD) and epilepsy is still not fully known so far. They were once regarded as genetic diseases [28,29,30]. However, some studies have reported that both genetic and environmental factors can contribute to these diseases [31,32,33]. Therefore, the objective of this study was to evaluate whether high levels of maternal exposure to NO_2_, SO_2_, and PM2.5 could increase the risk of neurological disorders such as autism spectrum disorder and epilepsy using the National Health Insurance claims data of South Korea.

## 2. Materials and Methods

### 2.1. Study Population

This study was approved by the Institutional Review Board of the Korea University Ansan Hospital (2021AS0317). Information obtained from the Korean National Health Insurance (NHI) claims database from January 2016 to December 2020 was used in this study. The NHI claims database provides information on all the insurance claims of the Korean population.

This study cohort included NHI claims data for babies with short gestation periods and low birth weights (“P07”), comprising singletons and twins (“Z38.0–Z38.5”) born in hospitals, and mothers who gave birth from January 2016 to December 2018. Multiple births and missing data were excluded, as they could have affected the results of this study.

The baseline characteristics, underlying diseases, and follow-up data of study subjects were extracted from the NHI claims database. The measured data on ambient air pollutants (PM2.5, CO, SO_2_, NO_2_, and O_3_) and heavy metals (Pb, Cd, Cr, Cu, Mn, Fe, Ni, and As) in South Korea from January 2015 to December 2018 were extracted from the Korea Environment Corporation (https://www.airkorea.or.kr/eng/ accessed on 18 April 2022.). The atmospheric conditions data were matched to mothers and their newborns based on the mother’s NHI registration area. Air pollutant measurement data measured during pregnancy were matched based on the mother’s health insurance claim registration area. The gestation period was divided into three stages. The first 1–3 months were defined as stage 1, 4–7 months as stage 2, and 8–10 months as stage 3. In the case of household income, the bottom 40% was defined as low and the top 5% was defined as high. Premature (“P07.2–F07.3”) and twin (Z38.3–Z38.5) codes were used.

Autism spectrum disorder (“F84.0–F84.9”), excepting Rett’s syndrome (F84.2), and epilepsy (“G40.0–G40.9”) were the disease codes used in this study. The minimum observation period for infants up to disease onset was maintained as two years or more.

### 2.2. Statistical Analysis

Data are presented as the mean ± standard deviation and number (%) of patients. The confounding variables and essential characteristics of the groups with and without autism spectrum disorder and groups with and without epilepsy were analyzed using an independent t-test for continuous variables and Fisher’s exact test or the chi-square test for categorical variables.

Logistic regression was performed, and odds ratios and 95% CIs for autism spectrum disorder or epilepsy adjusted for maternal age, education, infant sex, gestational season, and household income were analyzed for air pollutant and heavy metal exposure according to pregnancy stage in a single-pollutant model.

For autism spectrum disorder, two air pollutants and two heavy metals were selected according to forward selection with a four-pollutant model. After adjusting for maternal age, education level, infant sex, pregnancy period, and household income, the odds ratios and 95% CIs for autism spectrum disorder were analyzed based on exposure to air pollutants and heavy metals according to pregnancy stage.

For epilepsy, two air pollutants and three heavy metals were selected according to forward selection with a five-pollutant model. After adjusting for maternal age, education level, infant sex, pregnancy period, and household income, odds ratios and 95% CIs for epilepsy were analyzed based on exposure to air pollutants and heavy metals according to pregnancy stage. Additionally, adjusted odds ratios and 95% CIs for the neurologic disorders and exposure to air pollutants and heavy metals according to months of pregnancy in a one-pollutant model were obtained.

All statistical analyses were performed using SAS^®^ ver. 9.4 (Statistical Analysis Software 9.4, SAS Institute Inc., Cary, NC, USA). Differences were considered statistically significant if the *p*-value was less than 0.05.

## 3. Results

The medical records of 854,796 mothers from January 2016 to December 2020 and their newborns from January 2016 to December 2018 were reviewed (Figure 1). There were 5493 infants with autism spectrum disorder and 3190 infants with epilepsy. A total of 11,662 patients were excluded due to incomplete medical records. Finally, 843,134 patients were included in this study.

Table 1 shows the demographic and prenatal characteristics of the study subjects. For the infants with ASD, the proportion of maternal ages of 40 years or older was higher (23.5%) compared to that for infants without ASD (18.5%, *p* < 0.001). There were more males (71.6%) among the infants with ASD than the infants without ASD (51.25%, *p* < 0.001). Furthermore, more infants with ASD were born in winter (33.7%, *p* < 0.001), and these infants included more preterm babies (15.1%) and twins (3.5%, *p* < 0.001). For infants with epilepsy, the proportion of maternal ages over 40 was higher (23.9%) than that for infants without ASD (18.5%, *p* < 0.001) There were more males (55.3%) among the infants with epilepsy than the infants without epilepsy (51.3%, *p* < 0.001). Infants who were born in winter (36.7%) (*p* < 0.001) and those who were preterm (13.5%) were more prevalent among the infants with epilepsy than the infants without epilepsy (*p* < 0.001).

Table 2 shows the summary statistics of air pollutants and heavy metals by case. Figure 2 shows the spatial distribution of the mean PM2.5 and NO_2_ concentrations and the number of neurologic disorders in South Korea. An increase in the mean PM2.5 or Pb was associated with an increase in the incidence of ASD and epilepsy in newborns.

The adjusted odds ratios and 95% CIs for autism spectrum disorder and epilepsy and exposure to air pollutants and heavy metals according to the stage of pregnancy in the single-pollutant model are shown in Table 3.

The adjusted odds ratios and 95% CIs for autism spectrum disorder and exposure to air pollutants and heavy metals according to the stage of pregnancy in a four-pollutant model are shown in Table 4. The adjusted odds ratio and 95% CIs for epilepsy and exposure to air pollutants and heavy metals according to the stage of pregnancy in a five-pollutant model are shown in Table 5.

ASD was associated with increased total mean concentrations of SO_2_, NO_2_, and Pb during the entire pregnancy. SO_2_ (OR: 2.723, 95% CI: 1.971–3.761) and Pb (OR: 1.063, 95% CI: 1.019–1.11) were strongly associated with the incidence of ASD when infants were exposed to them in the third trimester of pregnancy. Epilepsy was associated with increased total mean concentrations of SO_2_ and NO_2_ during the entire pregnancy. Pb (OR: 1.109, 95% CI: 1.043–1.179) in the first trimester of pregnancy and Cd (OR: 2.193, 95% CI: 1.074–4.477) in the third trimester of pregnancy were also associated with the incidence of epilepsy.

Figure 3 shows the adjusted odds ratios and 95% CIs for the neurologic disorders and exposure to air pollutants and heavy metals according to months of pregnancy in a one-pollutant model. The association between the concentration of Pb by pregnancy period and the occurrence of ASD and epilepsy increased over the entire pregnancy period. On the other hand, Cd had a closer association with the occurrence of ASD and epilepsy in the early and late pregnancy periods.

## 4. Discussion

This study demonstrated that exposure to high concentrations of SO_2_, NO_2_, and Pb during the entire pregnancy period was associated with the development of ASD or epilepsy. The biological mechanism linking in utero exposure to air pollution and neural development in children is not yet fully understood. Maternal exposure to air pollution, including NO_2_/SO_2_ or PM2.5 and heavy metals, can lead to inflammation, oxidative stress, and DNA methylation placenta via lung tissues [34,35,36,37,38]. Oxygen and nutrient transport to the fetus are then disturbed, and inflammatory cytokines from the maternal circulation might be transported to the fetus, causing a fetal systemic inflammatory response which can adversely affect fetal development [34,35,36,39,40,41]. The fetal period is a critical window for brain development. Air pollutants in utero can significantly increase the susceptibility of infants to neurological diseases after birth [42].

This study demonstrated that exposure to high concentrations of SO_2_ during the third trimester was associated with the birth of children with ASD. This study also revealed that exposure to SO_2_ during the early and late pregnancy periods was associated with the birth of a child with epilepsy. Regarding ASD, a previous study showed no association between the level of prenatal SO_2_ exposure and the risk of ASD [43]. In that study, the average exposure level of SO_2_ during pregnancy was approximately 5.8 ppb during the observation period, which is similar to the result of our study (an average of 5 ppb), but the relationship between prenatal SO_2_ exposure and the risk of ASD showed a different result. Other studies have reported that prenatal SO_2_ exposure is associated with poor or impaired neurodevelopment in early childhood [44,45]. It is known that SO_2_ may lead to neurotoxicity by inducing oxidative stress, DNA damage, and apoptosis, as well as DNA methylation [46,47,48,49,50]. Previous studies on the absorption, distribution, and retention of SO_2_ in mammalian subjects have indicated that sulfur can be absorbed into the blood circulation and transported to the central nervous system [51,52]. When SO_2_ reaches the CNS, its biochemical effects can change the enzymatic activities of the CNS [53]. SO_2_ can depress some enzymatic activities of glucose metabolism [54]. With respect to prenatal exposure, Choi et al. [50] reported that prenatal DNA-methylation-associated SO_2_ exposure is associated with an increased ADHD rating scale in later childhood. Moreover, Liu et al. reported that prenatal SO_2_ exposure is positively related to the fetal hs-CRP level, a biomarker of systemic inflammation, and Liu et al. also suggested that prenatal SO_2_ exposure might interfere with fetal glucolipid metabolism by inducing fetal systemic inflammation [55]. However, the study reported no significant association between NO_2_ exposure and increased hs-CRP levels, indicating that the mechanisms through which SO_2_ and NO_2_ affect fetal neurodevelopment might be different. These are proposed mechanisms through which NO_2_ may interfere with neuronal development.

Regarding ASD, our results on prenatal NO_2_ exposure are consistent with the associations reported in previous studies [43,56]. These previous studies reported that during all pregnancy periods, prenatal NO_2_ exposure was associated with the risk of ASD. On the other hand, Gong et al. reported that prenatal NO_2_ exposure is not associated with the risk of ASD [57,58]. However, these studies reported that the average prenatal NO_2_ level of exposure was around 14 μg/m^3^ to 20 μg/m^3^ or 5.4 μg/m^3^ to 12.7 μg/m^3^ in the observation period. These levels of NO_2_ were found to be lower than those of studies showing an association between NO_2_ and ASD (Wang et al., 24 μg/m^3^, Volk et al., 32.24 μg/m^3^) or this study (46.20 μg/m^3^).

Regarding epilepsy, there have been studies reporting that postnatal NO_2_ exposure is associated with the risk of epilepsy [59,60]. However, to the best of our knowledge, this study is the first to demonstrate that prenatal NO_2_ exposure is associated with the risk of epilepsy. It is known that prenatal NO_2_ exposure can induce oxidative stress [61,62] and systemic inflammation [63]. Oxidative stress can induce the placenta to secrete factors detrimental to neurons and expose fetal brains to oxidative stress, thus adversely affecting neuronal development. Inflammation can expose a fetus to maternal immune activation and pro-inflammatory cytokines, which can adversely affect neurodevelopment [64,65,66,67]. In animal studies, it has been found that prenatal NO_2_ exposure can adversely affect neonatal behavioral development [68]. Michikawa et al. [69] suggested that NO_2_ affects the placenta by inducing inflammation, and this may be related to inflammation of the endometrium. Other studies have shown that maternal exposure to NO_2_ is significantly associated with placental DNA methylation levels known to affect fetal development [37] and with DNA methylations that are associated with apoptosis-related genes in cord blood cells [70]. These are proposed mechanisms through which NO_2_ may interfere with neuronal development.

This study demonstrated that high concentrations of lead (Pb) exposure during the late pregnancy period was associated with the birth of a child with ASD and that exposure to Pb during the early pregnancy period was associated with the birth of child with epilepsy. High concentrations of Pb have been observed in hair and nail samples from children with ASD [71,72]. Skogheim et al. [73], using maternal blood samples, also suggested that prenatal Pb exposure is associated with the risk of ASD. Regarding epilepsy, Sasmaz et al. [74] reported that Pb concentrations were significantly higher in the hair of epilepsy patients than in the healthy group. Other studies have reported that Pb exposure during early development is associated with cognitive deficits, as well as behavioral abnormalities [75]. In a zebrafish study, prenatal exposure to water-soluble fractions of Pb could induce autism-like behavior in larvae [76]. Chen et al. [77] reported that prenatal Pb exposure can induce neurobehavioral anomalies in mice. Microglia are the most important innate immune cells in the brain. Pb has been shown to activate inflammasome proteins associated with microglial activation [78] and trigger microglial activation, releasing inflammatory cytokines and neural apoptosis [79]. Pb is a neurotoxicant that can suppress brain plasticity in a critical period of neurodevelopment [80]. Prenatal Pb exposure can cross the placenta and accumulate in fetal tissues, threating the developing brain and adversely affecting placenta functions [81]. Pb has been associated with altered DNA methylation patterns, with some affected genes being related to neurodevelopment or cognitive function [82,83]. Pb can impact the brain through DNA methylation mechanisms as well as interactions with calcium-ion-dependent processes and oxidative damage [84].

This study demonstrated that exposure to high concentrations of cadmium (Cd) during the prenatal late pregnancy period was associated with the birth of a child with epilepsy. Cd might be released from the mother and transferred to the fetus via the placenta. High concentrations of Cd have been found in the hair of infants with mothers occupationally exposed to Cd [85]. Prenatal Cd exposure is known to affect infant growth and organ development [86]. In animal studies, it has been found that Cd can affect neural development [87,88]. Some studies have revealed that exposure to Cd in early pregnancy is related to cognition or ASD and ADHD [73,89]. However, Forns et al. [90] reported that prenatal exposure to Cd is not related to cognition.

There are several limitations of this study. First, ASD is known to develop from complex interactions between genetic and environmental risk factors [91]. However, genetic factors were not considered in the present study. Second, ASD and epilepsy are influenced by postnatal exposure to air pollution. However, the findings were not adjusted for postnatal exposure. Since the subjects of this study were infants, their exposure after birth was unlikely to have had a significant effect. Third, co-exposure to toxic metals has a synergistic effect. However, there was no adjustment for this effect. For example, Pb and mercury have been found to have synergistic negative effects on childhood cognitive ability and development [92]. Gorini et al. [93] also discussed the impacts of single-heavy-metal exposure and co-exposure to multiple metals on the development of ASD.

The strength of this study was that it demonstrated the association of prenatal exposure to heavy metals with ASD and epilepsy using air pollution data. Previous studies have studied the risk of prenatal exposure using the hair or nail of the child and the blood or hair of the mother. To the best of our knowledge, this study was the first to measure the prenatal risk of each heavy metal as an air pollutant. The results demonstrate a more direct association between heavy metals in air pollution and the risk of prenatal exposure to ASD and epilepsy.

## 5. Conclusions

The findings of this study suggest that exposure to SO_2_, NO_2_, and Pb during pregnancy can affect the development of neurologic disorders according to the timing of exposure, indicating that such exposure is related to fetal development. The relationships of ASD and epilepsy with air pollution identified in this study need to be further clarified through more personalized assessments and further epidemiological studies. In addition, research on the mechanisms of toxic substances is needed. All these efforts will further clarify the causal relationship between air pollution and the incidence of ASD and epilepsy.

## Figures and Tables

**Figure 1 medicina-59-00951-f001:**
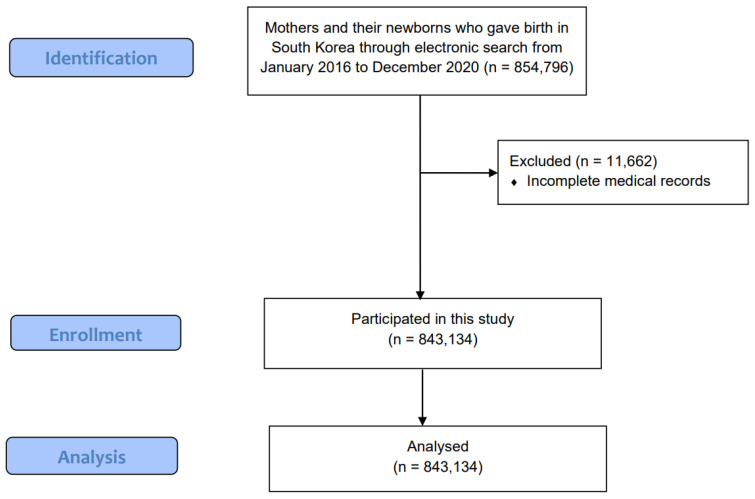
Consort flow diagram of this study.

**Figure 2 medicina-59-00951-f002:**
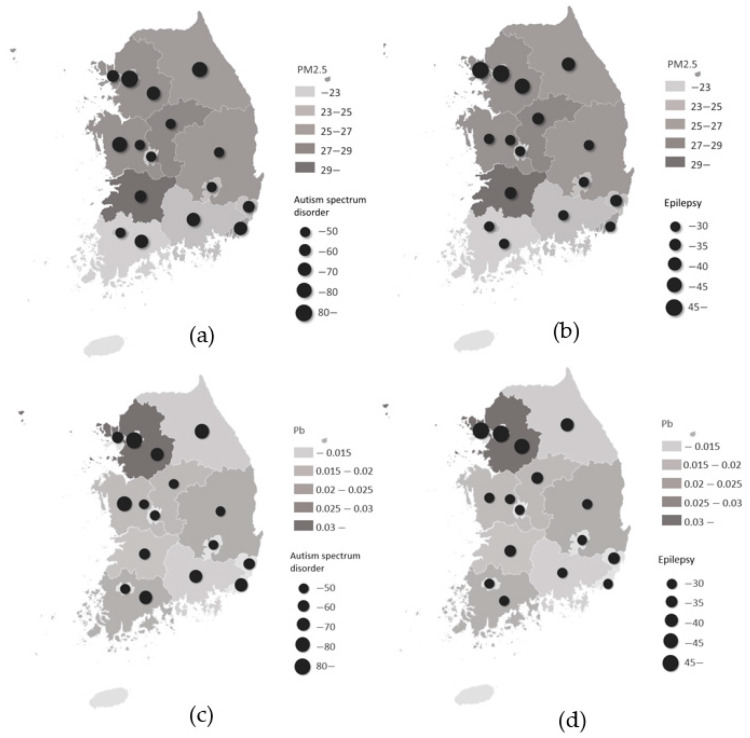
Spatial distribution of the mean PM2.5 and NO_2_ concentrations and the number of neurologic disorders in South Korea. (**a**) PM2.5 and autism spectrum disorder, (**b**) PM2.5 and epilepsy, (**c**) Pb and autism spectrum disorder, (**d**) Pb and epilepsy. Data are presented as the mean and number of patients.

**Figure 3 medicina-59-00951-f003:**
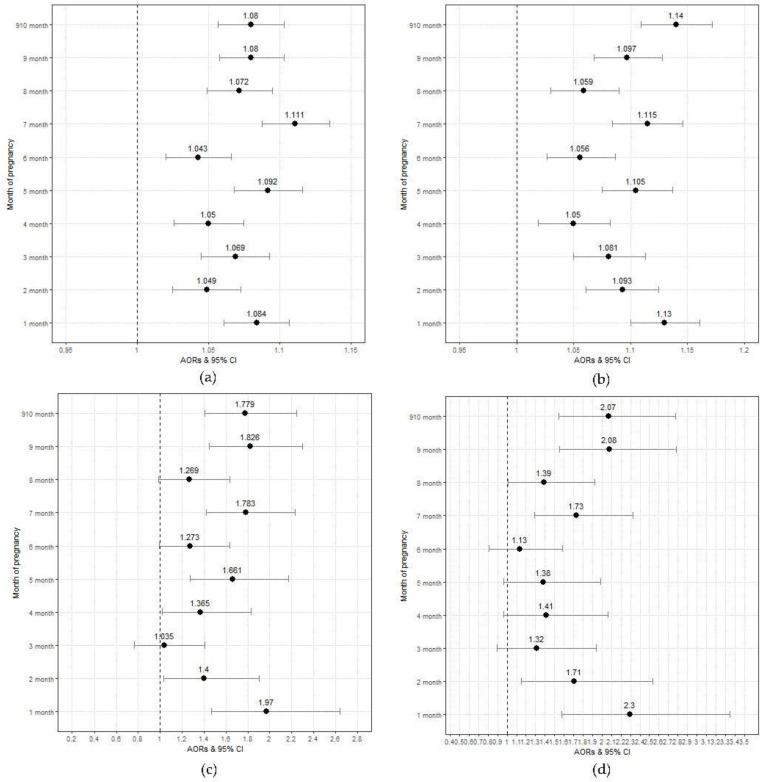
Adjusted odds ratios and 95% Cis for neurologic disorders and exposure to air pollutants and heavy metals by months of pregnancy in a one-pollutant model. (**a**) Pb and autism spectrum disorder by months of pregnancy, (**b**) Pb and epilepsy by months of pregnancy, (**c**) Cd and autism spectrum disorder by months of pregnancy, and (**d**) Cd and epilepsy by months of pregnancy. Logistic regression model adjusted for maternal age, education, infant sex, season of conception, and household income.

**Table 1 medicina-59-00951-t001:** Demographic and prenatal characteristics of the study subjects.

	Total	With Autism Spectrum Disorder	With Epilepsy
Maternal age (years) *†			
total	843,134	5493	3190
<20	68 (0.0)	0 (0.0)	1 (0.0)
20–30	70,501 (8.4)	460 (8.4)	246 (7.7)
30–40	616,720 (73.1)	3740 (68.1)	2182 (68.4)
40<	155,845 (18.5)	1293 (23.5)	761 (23.9)
Occupational status *			
Worked outside home	323,250 (38.3)	1945 (35.4)	1187 (37.2)
Household income *			
Low (40%)	292,469 (34.7)	1819 (33.1)	1095 (34.3)
Middle (40–95%)	531,455 (63.0)	3526 (64.2)	2006 (62.9)
High (95% <)	19,210 (2.3)	148 (2.7)	89 (2.8)
Infant sex *†			
Male	428,873 (51.2)	3934 (71.6)	1765 (55.3)
Season *†			
Winter	224,344 (26.6)	1850 (33.7)	1170 (36.7)
Spring	218,963 (26.0)	1332 (24.2)	716 (22.4)
Summer	204,840 (24.3)	1196 (21.8)	679 (21.3)
Fall	194,987 (23.1)	1115 (20.3)	625 (19.6)
Premature (27–36 weeks) *†			
<36 weeks	44,060 (5.3)	829 (15.1)	430 (13.5)
Multiple birth *			
twin (Z38.3–Z38.5)	18,734 (2.2)	193 (3.5)	77 (2.4)

Data are presented as the mean ± standard deviation and number (%) of patients. * *p* < 0.05 compared between groups with and without autism spectrum disorder. † *p* < 0.05 compared between groups with and without epilepsy.

**Table 2 medicina-59-00951-t002:** Summary statistics of air pollutants and heavy metals by case.

Air Pollutants and Heavy Metals	Mean	Median
PM10 (μg/m^3^)	45.35 ± 11.07	45.07 (36.67–52.68)
PM2.5 (μg/m^3^)	25.83 ± 6.17	25.33 (21.36–29.71)
SO_2_ (ppm))	0.005 ± 0.001	0.005 (0.004–0.005)
NO_2_ (ppm)	0.023 ± 0.007	0.023 (0.018–0.028)
O_3_ (ppm)	0.028 ± 0.010	0.028 (0.020–0.035)
CO (ppm)	0.493 ± 0.110	0.468 (0.412–0.575)
Pb (μg/m^3^)	0.024 ± 0.012	0.022 (0.015–0.033)
Cd (μg/m^3^)	0.001 ± 0.001	0.001 (0.001–0.001)
Cr (μg/m^3^)	0.005 ± 0.004	0.004 (0.002–0.006)
Cu (μg/m^3^)	0.025 ± 0.018	0.021 (0.013–0.034)
Mn (μg/m^3^)	0.032 ± 0.017	0.029 (0.020–0.040)
Fe (μg/m^3^)	0.672 ± 0.345	0.650 (0.429–0.838)
Ni (μg/m^3^)	0.005 ± 0.003	0.004 (0.003–0.006)
As (μg/m^3^)	0.004 ± 0.003	0.003 (0.002–0.005)

Data are presented as the mean ± standard deviation and median (25–75%).

**Table 3 medicina-59-00951-t003:** Adjusted a odds ratio and 95% CI of autism spectrum disorder or epilepsy and exposure to air pollutants and heavy metals according to the stage of pregnancy in a single-pollutant model.

		With Autism Spectrum Disorder	With Epilepsy
		OR (95% CI)	OR (95% CI)
PM10	1st stage	0.992 (0.988–0.996)	0.992 (0.987–0.998)
	2nd stage	1.023 (1.019–1.027)	1.021 (1.016–1.027)
	3rd stage	1.022 (1.018–1.026)	1.020 (1.015–1.026)
	total	1.023 (1.017–1.029)	1.023 (1.015–1.030)
PM2.5	1st stage	0.995 (0.989–1.003)	0.996 (0.987–1.005)
	2nd stage	1.036 (1.029–1.043)	1.033 (1.024–1.043)
	3rd stage	1.039 (1.031–1.046)	1.033 (1.024–1.043)
	total	1.048 (1.037–1.059)	1.047 (1.033–1.061)
SO_2_	1st stage	2.843 (2.240–3.609)	2.900 (2.129–3.950)
	2nd stage	4.149 (3.183–5.408)	3.443 (2.434–4.871)
	3rd stage	5.060 (3.819–6.703)	5.670 (3.920–8.202)
	total	6.060 (4.449–8.256)	6.039 (4.033–9.041)
NO_2_	1st stage	1.281 (1.222–1.343)	1.485 (1.396–1.580)
	2nd stage	1.463 (1.396–1.533)	1.587 (1.492–1.688)
	3rd stage	1.394 (1.329–1.462)	1.577 (1.481–1.680)
	total	1.439 (1.367–1.515)	1.669 (1.559–1.786)
O_3_	1st stage	0.719 (0.679–0.761)	0.679 (0.629–0.732)
	2nd stage	0.764 (0.723–0.807)	0.716 (0.666–0.770)
	3rd stage	0.862 (0.813–0.914)	0.653 (0.603–0.707)
	total	0.572 (0.522–0.626)	0.404 (0.358–0.457)
CO	1st stage	1.008 (1.004–1.012)	1.014 (1.009–1.019)
	2nd stage	1.022 (1.018–1.026)	1.025 (1.020–1.030)
	3rd stage	1.021 (1.017–1.025)	1.032 (1.026–1.037)
	total	1.027 (1.022–1.033)	1.040 (1.033–1.047)
Pb	1st stage	1.097 (1.069–1.126)	1.147 (1.108–1.186)
	2nd stage	1.088 (1.060–1.117)	1.101 (1.063–1.139)
	3rd stage	1.139 (1.110–1.169)	1.164 (1.125–1.205)
	total	1.143 (1.109–1.178)	1.181 (1.136–1.228)
Cd	1st stage	2.038 (1.350–3.076)	3.045 (1.789–5.181)
	2nd stage	1.874 (1.316–2.667)	1.750 (1.101–2.782)
	3rd stage	2.458 (1.815–3.328)	3.288 (2.263–4.777)
	total	3.287 (2.113–5.113)	5.389 (3.061–9.487)
Cr	1st stage	1.075 (0.980–1.180)	1.059 (0.936–1.197)
	2nd stage	1.121 (1.020–1.233)	1.054 (0.929–1.195)
	3rd stage	1.102 (0.995–1.221)	1.122 (0.982–1.282)
	total	1.153 (1.026–1.297)	1.112 (0.954–1.297)
Cu	1st stage	1.028 (1.010–1.046)	1.071 (1.047–1.094)
	2nd stage	1.045 (1.026–1.064)	1.088 (1.063–1.114)
	3rd stage	1.059 (1.039–1.078)	1.103 (1.077–1.129)
	total	1.052 (1.031–1.073)	1.109 (1.080–1.137)
Mn	1st stage	1.016 (0.997–1.036)	1.003 (0.978–1.029)
	2nd stage	1.042 (1.023–1.061)	1.023 (0.999–1.048)
	3rd stage	1.031 (1.012–1.051)	1.031 (1.005–1.056)
	total	1.036 (1.014–1.058)	1.024 (0.996–1.054)
Fe	1st stage	1.001 (1.000–1.002)	1.003 (1.001–1.004)
	2nd stage	1.004 (1.003–1.005)	1.005 (1.004–1.007)
	3rd stage	1.004 (1.003–1.005)	1.004 (1.003–1.006)
	total	1.004 (1.003–1.005)	1.006 (1.004–1.007)
Ni	1st stage	1.054 (0.945–1.176)	1.020 (0.883–1.177)
	2nd stage	1.202 (1.078–1.340)	1.191 (1.032–1.374)
	3rd stage	1.394 (1.248–1.558)	1.296 (1.120–1.501)
	total	1.302 (1.144–1.481)	1.248 (1.054–1.479)
As	1st stage	1.565 (1.390–1.763)	1.311 (1.118–1.537)
	2nd stage	1.912 (1.700–2.152)	1.481 (1.260–1.741)
	3rd stage	1.660 (1.488–1.852)	1.687 (1.461–1.947)
	total	2.622 (2.222–3.095)	2.039 (1.642–2.532)

Logistic regression model adjusted for maternal age, education, infant sex, season of conception, and household income.

**Table 4 medicina-59-00951-t004:** Adjusted odds ratios and 95% CIs for autism spectrum disorder and exposure to air pollutants and heavy metals according to the stage of pregnancy in a four-pollutant model.

		With Autism Spectrum Disorder
SO_2_ + NO_2_ + Pb + Cd		OR (95%CI)
SO_2_	Total	3.288 (2.306–4.687)
	1st stage	1.770 (1.338–2.342)
	2nd stage	2.128 (1.557–2.909)
	3rd stage	2.723 (1.971–3.761)
NO_2_	Total	1.322 (1.244–1.403)
	1st stage	1.233 (1.165–1.304)
	2nd stage	1.421 (1.346–1.501)
	3rd stage	1.267 (1.197–1.341)
Pb	Total	1.079 (1.017–1.145)
	1st stage	1.041 (0.993–1.092)
	2nd stage	0.980 (0.936–1.026)
	3rd stage	1.063 (1.019–1.110)
Cd	Total	0.441 (0.184–1.057)
	1st stage	1.233 (1.165–1.304)
	2nd stage	1.421 (1.346–1.501)
	3rd stage	1.267 (1.197–1.341)

Logistic regression model was adjusted for maternal age, education, infant sex, season of conception, and household income.

**Table 5 medicina-59-00951-t005:** Adjusted odds ratio and 95% CIs for epilepsy and exposure to air pollutants and heavy metals according to the stage of pregnancy in a five-pollutant model.

		With Epilepsy
SO_2_ + NO_2_ + Pb + Cd + As		OR (95%CI)
SO_2_	Total	3.702 (2.25–6.089)
	1st stage	2.106 (1.403–3.163)
	2nd stage	1.648 (1.061–2.560)
	3rd stage	2.897 (1.855–4.522)
NO_2_	Total	1.869 (1.696–2.059)
	1st stage	1.542 (1.422–1.672)
	2nd stage	1.680 (1.553–1.817)
	3rd stage	1.548 (1.427–1.681)
Pb	Total	1.064 (0.983–1.152)
	1st stage	1.109 (1.043–1.179)
	2nd stage	1.039 (0.976–1.107)
	3rd stage	1.031 (0.974–1.092)
Cd	Total	2.591 (0.738–9.102)
	1st stage	0.857 (0.328–2.243)
	2nd stage	0.757 (0.308–1.859)
	3rd stage	2.193 (1.074–4.477)
As	Total	0.273 (0.187–0.399)
	1st stage	0.461 (0.361–0.589)
	2nd stage	0.568 (0.436–0.741)
	3rd stage	0.613 (0.480–0.783)

Logistic regression model was adjusted for maternal age, education, infant sex, season of conception, and household income.

## Data Availability

Not applicable.

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
