# Peer review of "The Effect of Maternal Exposure to Air Pollutants and Heavy Metals during Pregnancy on the Risk of Neurological Disorders Using the National Health Insurance Claims Data of South Korea"

_medicina, 2023, doi:10.3390/medicina59050951_

Round 1

Reviewer 1 Report

Dear Authors,

The manuscript received for review with the title „Effect of maternal exposure to air pollutant and heavy metals during pregnancy on risk of neurological disorder using national health insurance claims data of South Korea” is of interest in terms of risks of autism spectrum disorder (ASD) and epilepsy using national health insurance claims data of South Korea. From my point of view, I recommend its publication in the form submitted, with small changes regarding the rules of technical editing of the journal, but also the English language check.

Congratulations to the authors.

Author Response

Reviewer1

Dear Authors,

The manuscript received for review with the title „Effect of maternal exposure to air pollutant and heavy metals during pregnancy on risk of neurological disorder using national health insurance claims data of South Korea” is of interest in terms of risks of autism spectrum disorder (ASD) and epilepsy using national health insurance claims data of South Korea. From my point of view, I recommend its publication in the form submitted, with small changes regarding the rules of technical editing of the journal, but also the English language check.

Congratulations to the authors.

Thank you for your reviewing. I spent much time for paper correction, then I have no time to check my English. Then, I will check my English next time

Reviewer 2 Report

Comments to author

The authors approached a significant investigation field Effect of Maternal Exposure to Air Pollutant and Heavy Metals during Pregnancy on Risk of Neurological Disorder using National Health Insurance Claims Data of South Korea”. The article can be interesting for the readers of the IJERPH and is suitable for publication in the journal after minor changes.

General Comments.

The authors tried to evaluate the  Effect of Maternal Exposure to Air Pollutant and Heavy Metals during Pregnancy on Risk of Neurological Disorder using National Health Insurance Claims Data of South Korea” however, introduction section need some improvement. Authors should provide some more detail about the current situation of  Air Pollutant and Heavy Metals in South Korea and its overall situation globally. Add some more citations from the recent literature with new, pertinent and relevant information on the studied topic.

Material and Methods

Well written

Discussion.

Check the length of the sentences and try to avoid long sentences until and unless it is needed.

References.

1.     Authors have cited only few studies upto or after 2020. Please add some more citations from most recent literature to strengthen the results. 

Author Response

Reviewer2

The authors approached a significant investigation field “Effect of Maternal Exposure to Air Pollutant and Heavy Metals during Pregnancy on Risk of Neurological Disorder using National Health Insurance Claims Data of South Korea”. The article can be interesting for the readers of the IJERPH and is suitable for publication in the journal after minor changes.

General Comments.

The authors tried to evaluate the  Effect of Maternal Exposure to Air Pollutant and Heavy Metals during Pregnancy on Risk of Neurological Disorder using National Health Insurance Claims Data of South Korea” however, introduction section need some improvement. Authors should provide some more detail about the current situation of  Air Pollutant and Heavy Metals in South Korea and its overall situation globally. Add some more citations from the recent literature with new, pertinent and relevant information on the studied topic.

→ We appreciate your comment. We revised the introduction in more detail on the overall situation of global air pollution and the situation of air pollutants in Korea by referring to recent literature as follows.

“Air pollution is recognized as a major risk factor for global human health. Exposure to air pollution has been linked to increased mortality and morbidity, contributing significantly to the overall burden of disease globally. The number of all-causes deaths from overall air pollution increased by 2.62% globally from 1990 to 2019. The Health Effects Institute’s State of Global Air reported that PM2.5, a type of fine particulate air pollution, is the sixth highest risk factor for global deaths, accounting for almost four million deaths in 2019 alone. Over 90% of the world’s population in areas where the air quality standards set by the WHO are not being met. Especially in Asian megacities, the air pollution concentrations have been observed to be the highest in the world. Korea has undergone rapid economic growth in recent decades, and the quality of the air has worsened.

Air pollution is caused by complex components including nitrogen dioxide (NO2), sulfur dioxide (SO2), and particulate matter (PM). NO2, SO2 and PM and indirectly generated so for ozone (O3) are major air pollutants that are related to exhaust from vehicles or industrial energy consumption, which are in turn caused by urbanization or industrialization.

Heavy metals are generally amalgamated with PM. Heavy metals mainly originate from diesel and gasoline exhaust fumes from local traffic and industrial areas.”

Material and Methods

Well written

Discussion.

Check the length of the sentences and try to avoid long sentences until and unless it is needed.

References.

  1. Authors have cited only few studies upto or after 2020. Please add some more citations from most recent literatureto strengthen the results. 

→ Thank you for your comment. We add some citations from most recent literature (38, 50, 62, 70, 78, 79) and revised the discussion as follows to strengthen the results.

“It is known that SO2 may lead to neurotoxicity by inducing oxidative stress, DNA damage, and apoptosis, DNA methylation”

“About prenatal exposure, Choi et al. have reported that prenatal SO2 exposure associated with DNA methylation are associated with increases in ADHD rating scale in later childhood. And…”

“Other studies have shown that maternal exposure to NO2 is significantly associated with placenta DNA methylation levels known to affect fetal development, as well as with DNA methylation that are associated with gene related to apoptosis in cord blood cells.”

“Microglia are considered to be the most important innate immune cells in the brain. Pb is shown to activate inflammasomes protein associated with microglial activation. Pb has been shown to trigger microglial activation, resulting in the release of inflammatory cytokines and neural apoptosis.”

Reviewer 3 Report

               The paper "Effect of Maternal Exposure to Air Pollutant and Heavy Metals during Pregnancy on Risk of Neurological Disorder using National Health Insurance Claims Data of South Korea" presented to me for evaluation covers the issue of the relationship between air pollution in which pregnant women live and the risk of neurological disorders in their children. The data was obtained by the Authors at National Health Insurance Claims Data of South Korea and Korea Environment Corporation.

               The relationship between the content of PM2.5, CO, SO2, NO2, and O3 as well as Pb, Cd, Cr, Cu, Mn, Fe, Ni, and As in the atmospheric air and the health condition of children was analyzed. All air quality data and the National Health Insurance Claims Data of South Korea were collected between 2015 and 2020.

               The comparisons made show the relationship between air pollution and the occurrence of neurological disorders in children. Research is valuable from the point of view of assessing air quality and exposure of women and their children to pollution.

               The results of these studies have been presented in figures and in clear tables, which makes the work easier to read. The analysis of data for South Korea may in the future be a reference for further research in this area covering other diseases in people of different ages.

               In general, the work is valuable and, after making corrections, it can be the subject of further processing.

               My comments below:

               In the title of the work, the Authors write: "……Air Pollutant and Heavy Metals….", and are heavy metals not air pollutants? Maybe it's worth modifying the title a bit and adding dusts and C, S and N oxides? Please consider it.

               In the introduction, the authors should add a short fragment about the sources of origin of the discussed air pollutants and specify whether these are natural or artificial (anthropogenic) sources. The background of these compounds in the air must also be given.

               There is no mention of heavy metals and O3 in the introduction, only information about PM2.5, CO, SO2, NO2.

[5] – there should be a comma after 1 and 2

[37, 39, 41, 43, 49, 52, 53, 55, 56, 211, 214, 216, 221, 226, 227, 230, 232, 235, 241, 244, 253, 261, 268, 271, 272 , 274, 275, 276, 278, 280, 282, 287, 288, 289, 290, 291, 293, 300, 301] - references to literature should be placed before a full stop at the end of a sentence, not after a full stop

[51] - the abbreviations BDNF and CREB need to be explained, it will be clearer for the recipient

[54] – the abbreviation ASD needs to be explained not only in the abstract, but also in the introduction

[145] - table 2 - three decimal places at PM10 and PM2.5 in the mean and median are not needed. Two places are enough. Similarly for the standard deviation SD in the whole table 2. No spaces after O3.

[177 – 178] - Table 5 - should be With epilepsy, not With Epilepsy

[191 - 192] - Figure 3 - the font is not legible, but here the decision must be made by the editors whether it will be legible for the recipient

Figure 3 should be below text [199 - 204], not above

[247 - 248] - should be µg/m3, not µg/m3

[256] - no full stop at the end of the sentence

[258, 268, 269, 274, 300] - enter the reference number of the bibliography after the author's name

[333 - 517], the entire bibliography should be prepared according to the editorial requirements:

• publication year should be written in bold.

• name of the journal should be written in italics

• volume of the journal should be written in italics.

Author Response

Reviewer3

     The paper "Effect of Maternal Exposure to Air Pollutant and Heavy Metals during Pregnancy on Risk of Neurological Disorder using National Health Insurance Claims Data of South Korea" presented to me for evaluation covers the issue of the relationship between air pollution in which pregnant women live and the risk of neurological disorders in their children. The data was obtained by the Authors at National Health Insurance Claims Data of South Korea and Korea Environment Corporation.

               The relationship between the content of PM2.5, CO, SO2, NO2, and O3 as well as Pb, Cd, Cr, Cu, Mn, Fe, Ni, and As in the atmospheric air and the health condition of children was analyzed. All air quality data and the National Health Insurance Claims Data of South Korea were collected between 2015 and 2020.

               The comparisons made show the relationship between air pollution and the occurrence of neurological disorders in children. Research is valuable from the point of view of assessing air quality and exposure of women and their children to pollution.

               The results of these studies have been presented in figures and in clear tables, which makes the work easier to read. The analysis of data for South Korea may in the future be a reference for further research in this area covering other diseases in people of different ages.

               In general, the work is valuable and, after making corrections, it can be the subject of further processing.

               My comments below:

               In the title of the work, the Authors write: "……Air Pollutant and Heavy Metals….", and are heavy metals not air pollutants? Maybe it's worth modifying the title a bit and adding dusts and C, S and N oxides? Please consider it.

→ Thank you for your advice. I correct the title.

               In the introduction, the authors should add a short fragment about the sources of origin of the discussed air pollutants and specify whether these are natural or artificial (anthropogenic) sources. The background of these compounds in the air must also be given.

→ Thank you for your correction. We added a sentence about the sources of air pollutants in introduction as follows.

“Exhaust from vehicles or heating and cooling in buildings are sources of air pollutants such as NO2, SO2 and PM, and indirectly so for ozone (O3). Heavy metals are gener-ally amalgamated with PM. Heavy metals mainly originate from diesel and gasoline exhaust fumes from local traffic and industrial areas.”

               There is no mention of heavy metals and O3 in the introduction, only information about PM2.5, CO, SO2, NO2.

→ We appreciate for your comment. We added a sentence about heavy metals and O3 in the introduction as follows.

“Exhaust from vehicles or heating and cooling in buildings are sources of air pollutants such as NO2, SO2 and PM, and indirectly so for ozone (O3). Heavy metals are gener-ally amalgamated with PM. Heavy metals mainly originate from diesel and gasoline exhaust fumes from local traffic and industrial areas.””

[5] – there should be a comma after 1 and 2

→ Thank you for your correction. We added a comma after 1 and 2 as follows.

“Kuen Su Lee1,†, Won Kee Min2,†

[37, 39, 41, 43, 49, 52, 53, 55, 56, 211, 214, 216, 221, 226, 227, 230, 232, 235, 241, 244, 253, 261, 268, 271, 272 , 274, 275, 276, 278, 280, 282, 287, 288, 289, 290, 291, 293, 300, 301] - references to literature should be placed before a full stop at the end of a sentence, not after a full stop

→ We appreciate for your correction. We have revised all references to literature be placed before a full stop at the end of a sentence.

[51] - the abbreviations BDNF and CREB need to be explained, it will be clearer for the recipient

→ Thank you for your comment. We revised that paragraph as follows.

“Regarding neurodevelopment, it is known that PM2.5 can induce oxidative stress and inflammatory response and that both oxidative stress and inflammatory response can affect expression levels of brain-derived neurotrophic factor (BDNF) and cyclic AMP-response element-binding protein (CREB) known to be neurodevelopment factors.”

[54] – the abbreviation ASD needs to be explained not only in the abstract, but also in the introduction

→ We appreciate for your correction. We explained the abbreviation ASD in the introduction as follows.

“The etiology of autism spectrum disorder (ASD) and epilepsy is still not fully known so far.”

[145] - table 2 - three decimal places at PM10 and PM2.5 in the mean and median are not needed. Two places are enough. Similarly for the standard deviation SD in the whole table 2. No spaces after O3.

→ Thank you for your correction. We revised the mean, median and standard deviation SD two decimal places. And we added a space after O3.

[177 – 178] - Table 5 - should be With epilepsy, not With Epilepsy

→ We appreciate for your correction. We revised it with “With epilepsy”.

[191 - 192] - Figure 3 - the font is not legible, but here the decision must be made by the editors whether it will be legible for the recipient

→ Thank you for your comment. We look forward to hearing from the editor. We are willing to provide higher quality Figrue if the editor wishes.

Figure 3 should be below text [199 - 204], not above

→ The picture position has been corrected.

[247 - 248] - should be µg/m3, not µg/m3

→ We appreciate for your correction. We revised it with “µg/m3”.

[256] - no full stop at the end of the sentence

→ Thank you for your correction. We added full stop at the end of the sentence.

[258, 268, 269, 274, 300] - enter the reference number of the bibliography after the author's name

→ We appreciate for your correction. We revised enter the reference number of the bibliography after the author's name

[333 - 517], the entire bibliography should be prepared according to the editorial requirements:

  • publication year should be written in bold.
  • name of the journal should be written in italics
  • volume of the journal should be written in italics.

→ Thank you for your correction. We revised the entire bibliography according to the editorial requirement.